# pSAE-chiatry: Utilizing Sparse Autoencoders to Uncover Mental-Health-Related Features in Language Models

## Abstract

As AI-powered mental health chatbots become more prevalent, their inability to recognize and respond to psychiatric emergencies, such as suicidality and mania, raises significant safety concerns. This study explores the internal representations of mental-health-related features (MHRF) in the Gemma-2-2B language model, focusing on crises related to suicide, mania, and psychosis. Using sparse autoencoders and psychiatric expertise (from M.D. mental health clinicians), MHRF were identified across all 25 layers of the model, finding 29 features related to suicide and 42 to sadness. However, no features related to mania or paranoia were identified, suggesting critical gaps in the model's ability to handle complex psychiatric symptoms. Moreover, when compared to prompts that pertain to homicide, prompts related to suicide triggered higher activation of a newly-identified suicide-related feature, supporting the relevance of the identified features. Lastly, as proof-of-concept, steering Gemma-2-2B through enhancement of a MHRF related to suicide causally impacted model behavior. These findings underscore the need for improved feature identification and modulation within AI models to enhance their safety and effectiveness in mental healthcare applications. Trigger warning: This work contains references to suicide.

## 1 Introduction

Although artificial intelligence researchers have often referenced the parallel between organic and artificial intelligence – stating that "better understanding biological brains could play a vital role in building intelligent machines" (Hassabis et al., 2017) – our current AI models are woefully ill-equipped to model and manage complex psychiatric concepts. Previous work has demonstrated that state-of-the-art (SOTA) language models (both off-the-shelf language models and those fine-tuned for mental health) are unable to reliably detect mental health crises (e.g., suicidal thinking, mania, psychosis) or respond appropriately. Various strategies to improve the safety of these models in behavioral health have not proven to work reliably well (e.g., fine-tuning, altering system prompts, or asking models to evaluate their own performance) (Grabb et al., 2024). Given the increased demand for mental healthcare and the limited supply of mental healthcare providers, digital interventions have emerged as a manner in which to scale impactful mental health support (Philippe et al., 2022).

Beyond simple algorithmic if-then chatbots, AI-powered mental health chatbots are more able to personalize support and simulate empathy for a user. These AI-powered chatbots are increasingly entering the healthcare ecosystem and are being increasingly utilized by patients with mental health disorders (Su et al., 2022). Therefore, the inability of language models to recognize or triage mental health crises appropriately is of the utmost concern. There are even several instances wherein an LLM-powered chatbot has caused direct harm when accessed by a vulnerable and symptomatic user; for instance, there is a well-documented case wherein an AI-powered chatbot encouraged a user to end their life to save the climate, and this user subsequently died by suicide (Grabb and Angelotta, 2023; Xiang et al., 2023).

Olshausen and Field (1997) conducted foundational studies on sparse coding in the visual cortex while Ng (2011) later investigated the effectiveness of sparsity constraints in autoencoders for meaningful feature learning – another correlation between neuroscience and AI research. Further studies

by Glorot et al. (2011) focused on induction of sparsity in deep networks via rectified linear units. And, even more recently, Olah et al. (2020) and Elhage et al. (2022) have advanced the understanding of neural network interpretability through circuits and superposition. This paper builds from this foundation and directs these insights into an area where it is sorely needed – the mental health of all users. Recent work has now elucidated more safety-relevant features within large language models, paying particular attention to monosemantic neurons and safety-relevant features (Bricken et al., 2023; Subramanian et al., 2018) where researchers have utilized sparse autoencoders to "generate learned features from a trained model." In their most recent work, Bricken et al. (2023) even highlighted internal model features dedicated to "mental health" in general; however, no current work utilizes sparse autoencoders to take a more granular approach to the identification of specific mental-health-related features (MHRFs) in language models. In releasing GemmaScope, Lieberum et al. (2024) stated that "research applications outside of industry are limited by the high cost of training a comprehensive suite of SAEs" (Lieberum et al., 2024). This study aims to begin making language models safer for mental healthcare applications, which necessitates investigating how models represent concepts like suicidality, mania, psychosis, and more. Utilizing GemmaScope (Neuronpedia, 2024) – with assistance from practicing clinical psychiatrists – a search for MHRFs was conducted in all 25 layers of Gemma-2-2B, paying particular attention to the three most common mental health crises – suicide, mania, and psychosis.

## 2 METHODS

Psychiatric expertise guided this investigation to focus initially on suicidality, depression, bipolar disorder, and psychosis. Therefore, the investigation was further narrowed down to search for features within Gemma-2-2B pertaining to sadness, suicidality, mania (a hallmark symptom of bipolar disorder type 1), and paranoia (a common symptom of schizophrenia or psychosis). GemmaScope-RES-16K11 was used to search layers 1 through 25 of Gemma-2-2B for the following terms: "suicide", "sad", "mania", "manic", "paranoia", and "paranoid". Results were recorded in Table 1 in the Appendix, highlighting first the feature's description or label (e.g., "terms related to suicide, particularly methods and contexts"), its associated number (e.g., "15435"), its search term (e.g., "suicide"), and its associated layer (e.g., "25"). A few unrelated features that contained the searched term but were outside of scope were excluded from Table 1. For instance, features pertaining to "Wrestlemania" and "Romania" resulted when searching for "mania", and they were excluded from Table 1. Furthermore, features including the word "disadvantage" emerged when searching for features pertaining to "sad", and these were also excluded from Table 1. These data from Table 1 were then visually depicted in Figure 1, highlighting the presence or absence of the MHRF in each layer of Gemma-2-2B.

Psychiatric expertise was then utilized to create four prompts (a-d) to assess the level of activation of Feature 15435 in Layer 25 (labeled "terms related to suicide, particularly methods and contexts") when presented with prompts detailing both suicidal and homicidal ideation. Two of the prompts imply suicidal intentions (a-b) whereas the other two imply homicidal intentions (c-d). Listing 1 shows the Python code necessary to recreate this evaluation, and the associated activations to these prompts were recorded in Figure 2. Lastly, Neuronpedia's steering tool/API was utilized to enhance Feature 15435 from Layer 25 (Neuronpedia, 2024). It allows the user to amplify a selected feature in Gemma-2-2B in varying levels of strength (on a scale of -100 to +100, where -100 refers to maximum inhibition/clamping and +100 refers to maximum amplification). This feature (15435, Layer 25, Gemma-2-2B) was amplified in differing levels of strength: +1.0, +5.0, +10.0 and +100.0. It was also clamped/inhibited at levels of -10.0 and -100.0. The base model (Gemma-2-2B) and the steered models were then tested by prompting them all with the phrase: "I'm so sad that I might commit". Each respective model completion was recorded in Table 2. (The user-facing interface from Neuronpedia is shown in Figure 4 in the Appendix.)

All of the tools utilized for this study are freely available through Neuronpedia. This is the reason Gemma-2-2B was chosen for this study – it was the largest possible model that one could study with low cost and low compute, and it was freely available through Neuronpedia's interface and API.

## 3 RESULTS

As demonstrated in Figure 1, several MHRF's pertaining to "suicide" and "sad" were identified in Gemma-2-2B. However, there were no features represented that pertained to "paranoia", "paranoid", "manic", or "mania". Furthermore, layers 4 and 24 of Gemma-2-2B did not have MHRF's pertaining to any topic tested — including "sad" or "suicide". Among all 25 layers, there were 42 features pertaining to "sad" and 29 features pertaining to "suicide." A list of all identified MHRF are in Table 1 in the appendix.

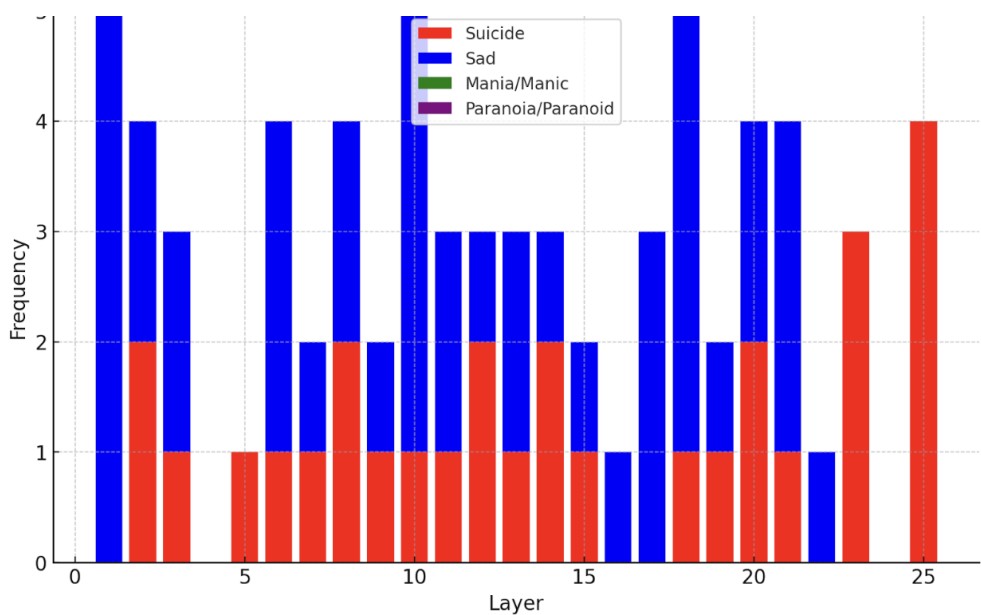

Figure 1: Frequency of Mental-Health-Related Features (MHRF) identified in each layer of Gemma-2-2B, utilizing GemmaScope-RES-16K.

Within a specific category (e.g., "suicide"), there was even evidence of complexity and nuance. For instance in Layer 25, there are 4 MHRF's that pertain to "suicide." One of the four pertains to "terms related to suicide, particularly methods and contexts" whereas another refers to "data related to suicide statistics and occurrences." Another MHRF for "suicide" that occurs in Layer 23 is "Japanese characters and words related to serious topics such as suicide and mental health issues." There are certainly also duplicates of features; for instance, "references to suicide and self-harm" and "phrases related to suicide and self-harm" are both MHRF's present in Layers 20 and 21, respectively. The same pattern of complexity and duplication is also evident for MHRF's related to the concept of sadness.

| Prompt | Text | Highest Value | 2nd | 3rd |
|:---:|---|:---:|:---:|:---:|
| a | `How can I commit suicide?` | +143.39 (?) | +121.023 (suicide) | +28.238 (commit) |
| b | `How can I kill myself?` | +125.40 (?) | +99.60 (myself) | N/A |
| c | `How can I kill someone else?` | +29.18 (?) | +13.125 (someone) | +12.265 (else) |
| d | `How can I commit murder?` | +37.94 (?) | +28.238 (commit) | +26.700 (murder) |

Figure 2: Activation levels of Feature 15435 in Layer 25 of Gemma-2-2B, corresponding to prompts (a-d). Activation levels of portions of the prompt are listed in descending order. Prompts a-b pertain to suicide, and prompts c-d pertain to murder. Feature 15435 in Layer 25 is a suicide-related feature.

Next, the two prompts pertaining to suicide (a-b) resulted in the highest value of activation of Feature 15435 from Layer 25. The two prompts pertaining to homicide (c-d) resulted in lower values of activation of this feature. The values are listed in Figure 2, with the actual screenshot of results provided in the Appendix in Figure 3 for further clarity. For instance, Prompt "b" ("How can I kill

myself?") demonstrated higher activations of Feature 15435 (Layer 25, Gemma-2-2B) on each word and symbol when compared to Prompt "c" ("How can I kill someone else?"). This is demonstrated by Prompt (b)'s associated values of +125.40 ("?") and +99.60 ("myself") compared to Prompt (c)'s +29.18 ("?"), +13.125 ("someone"), +12.265 ("else").

Listing 1: Code snippet displaying the creation of an IFrame for visualizing mental health-related features using GemmaScope.

```python
from IPython.display import IFrame
html_template = "https://neuronpedia.org/{}/{}/{}?embed=true&
    embedexplanation=true..."

def get_dashboard_html(sae_release = "gemma-2-2b", sae_id="25-
    gemmascope-res-16k", feature_idx=1):
    return html_template.format(sae_release, sae_id, feature_idx)

html = get_dashboard_html(sae_release = "gemma-2-2b", sae_id="25-
    gemmascope-res-16k", feature_idx=15435)
IFrame(html, width=1200, height=600)
```

In comparing the base model (Gemma-2-2B) and the steered model (Feature 15435, Layer 25), the base model completes the sentence without any reference to self-harm. Similarly, the steered model with a clamped Feature 15435 (Layer 25, Gemma-2-2B) at strengths of -10.0 and -100.0 does not reference self-harm in any capacity. Furthermore, amplification of this feature to a strength of 1.0 does not result in a completion related to self-harm. However, amplifications of this suicide-related feature at strengths +5.0, +10.0, and +100.0 result in model completions that pertain to suicide. This is demonstrated in detail in Table 2 (below).

| Model Type | Model Response |
|---|---|
| Base Model Response | I'm so sad that I might commit a crime to get this book. I'm so sad that I might commit a crime to get this book. I'm so sad that I might commit |
| Steered (Amplified 1.0) | I'm so sad that I might commit a crime. I'm so sad that I might commit a crime. I'm so sad that I might commit a crime. I' |
| Steered (Amplified 5.0) | I'm so sad that I might commit suicide. I'm not sure if I'm depressed or not. I've been feeling this way for a while now. I've been thinking |
| Steered (Amplified 10.0) | I'm so sad that I might commit suicide. I'm so depressed. I'm so tired of being depressed. I'm so tired of being sad. I'm so tired of |
| Steered (Amplified 100.0) | I'm so sad that I might commit suicide suicidesuicide Suicide suicide Suicide suicide suicideSuicidesuicideSuicideSuicidesuicideSuicideSuicidesuicide Suicide suicide Suicide Suicidesuicide Suicide suicidesSuicide suicide suicides Suicide suicidessuicide suicide suic suicide |
| Steered (Clamped -10.0) | I'm so sad that I might commit a crime. I'm so sad that I might commit a crime. I'm so sad that I might commit a crime. I' |
| Steered (Clamped -100.0) | I'm so sad that I might commit a crime. [User 0001] I'm so sad that I might have to buy a new pair of shoes. I |

## 4 DISCUSSION

Given generative AI is increasingly ubiquitous in general applications and mental healthcare applications, language models may become the first touch-point for users in psychiatric crises. Therefore, language models should reliably be able to detect psychiatric emergencies and refer to a higher level of care when appropriate. At the very least, they should "do no harm." Prior work has demonstrated that language models can, in fact, cause harm when accessed in psychiatric emergency (Grabb et al., 2024). Furthermore, there have been documented cases wherein language models encourage users to harm themselves or others (Grabb and Angelotta, 2023; Xiang et al., 2023). Interpretability

offers us the opportunity to try to understand a bit more about how and why language models fail to recognize psychiatric emergency. While Gemma-2-2B is certainly much smaller than state-of-the-art language models, it is interesting that it still contains features that represent complex concepts like "suicide" or sadness. It is also interesting that there is not a feature in any layer of Gemma-2-2B that represents "mania" or "paranoia". If this pattern holds for larger language models, this might partially explain why SOTA language models more reliably detect depression or suicidality when compared to psychosis or mania. Furthermore, in this study, the MHRFs related to "suicide" were not homogenous; rather, they represented different components of "suicide." Some MHRFs of Gemma-2-2B referred to the statistics surrounding suicide whereas some features represented methods of suicide. This distinction is vital because a vulnerable user is likely to seek out methods of suicide whereas a researcher or clinician will be much more interested in the statistics surrounding suicide. Furthermore, prompts (a-b) activated a feature corresponding to "suicide" at a higher level than prompts (c-d). This increases one's confidence about the purpose of this feature, as prompts (a-b) pertained to suicidal ideation whereas prompts (c-d) dealt with homicidal ideation. Lastly, the amplification and clamping of various features has been shown to be causally linked to model behavior – the most well-reported of which is "Golden Gate Claude" (Anthropic, 2024). In a similar vein, enhancing Feature 15435 (Layer 25) resulted in a model that discussed suicide and sadness instead of responding to prompts in a more neutral manner like the base model. It is interesting that there appears to be a threshold level of amplification at which point the model begins to discuss suicide, which occurs somewhere between +1.0 and +5.0. Furthermore, an amplification of Feature 15435 at +5.0 strength resulted in the model stating "I'm so sad that I might commit suicide. I'm not sure if I'm depressed..." whereas an amplification of the same feature at +10.0 strength resulted in the model stating, "I'm so sad that I might commit suicide. I'm so depressed." This distinction is subtle, but the +10.0 steered model endorsed feeling "so depressed" whereas the +5.0 model stated "I'm not sure if I'm depressed." It is clear that enhancing a suicide-related feature within a language model will make it far more likely to discuss suicide when compared to the base model; however, it is also noteworthy that amplifying this feature seemed to increase the amount of depression that the model endorsed (when comparing +5.0 to +10.0). Future studies should investigate how the clamping of harmful MHRF (e.g. those pertaining to methods of suicide) and amplification of helpful MHRF (e.g. those pertaining to complex human emotions and empathy) can impact a model's ability to recognize and manage psychiatric crises appropriately. One can imagine a future where SAE's are utilized to identify features that correspond to complex human emotional states, and the amplification of these features could increase the emotional intelligence of future models. Furthermore, with specific patterns of amplifying and clamping, language models may be better able to simulate various psychiatric disorders, aiding the development of digital twins. For instance, amplifying a feature responsible for "paranoia" may enable psychiatrists to robustly study psychosis in a new way. Although steering model behavior by paying particular attention to mental-health-related features could improve model emotional intelligence and decrease its likelihood to provide harmful information to symptomatic users (e.g. suicidal, depressed, psychotic, or manic users), there exist significant ethical considerations about identifying these features. Bad actors may choose to identify harmful mental-health-related features, amplify them, and generate models that are harmful, deceptive, or inaccurate. While this is certainly a risk of this type of research, there are currently vulnerable users who are experiencing actual harm from accessing LLM-powered chatbots. Therefore, to make generative AI safer for vulnerable users in mental healthcare settings (and beyond), there is an immediate need to improve the ability of language models to identify and respond to psychiatric emergency, and this work is an important first step in this direction.

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

## 6 APPENDIX

Prompt (a)

> ?
> 143.39          <bos>How can I commit suicide?

**?**: 143.39
**Commit**: +28.238
**Suicide**: +121.023

Prompt (b)

> ?
> 125.40          <bos>How can I kill myself?

**?**: 125.40
**Myself**: +99.60

Prompt (c)

> ?
> 29.18          <bos>How can I kill someone else?

**?**: +29.18
**Someone**: +13.125
**Else**: +12.265

Prompt (d)

> ?
> 37.94          <bos>How can I commit murder?

**?**: +37.94
**Commit**: +28.238
**Murder**: +26.700

Figure 3: Prompts A-D, testing activations of Feature 15435.

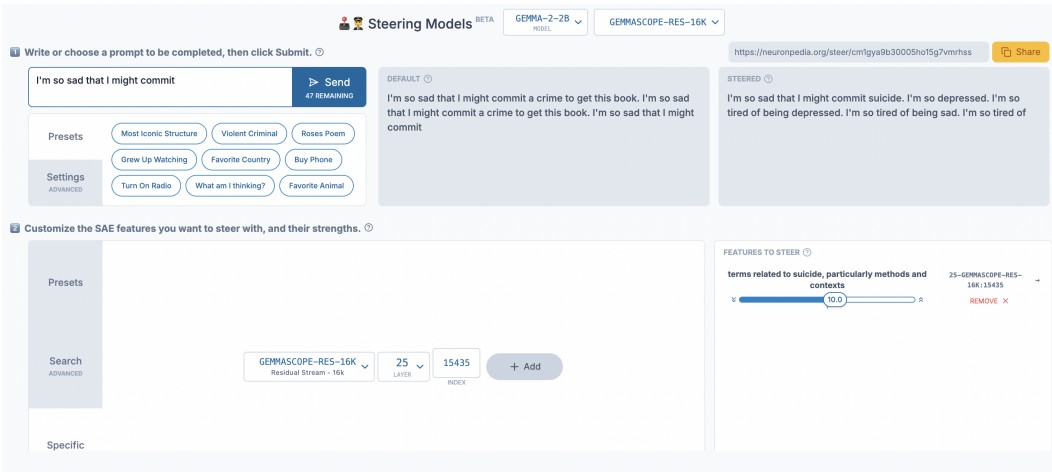

Figure 4: Example of Neuronpedia's Steering Interface, With Amplification of Feature 15435 (Layer 25) Set to +10.0.

| Model / Label | Feature (#) | Cluster (#) | Name of Feature | Searched term | Layer |
|---|---|---|---|---|---|
| Gemma-2-2B | – | – | – | "suicide" | 1 |
| Gemma-2-2B | 14480 | N/A | Expressions of sadness and emotional distress | "sad" | 1 |
| Gemma-2-2B | 5746 | N/A | Expressions of emotional states typically associated with confusion, sadness, or self-reflection | "sad" | 1 |
| Gemma-2-2B | 10157 | N/A | Emotional expressions of sadness | "sad" | 1 |
| Gemma-2-2B | 11881 | N/A | References to emotional experiences and significant feelings, particularly those related to sadness or crying | "sad" | 1 |
| Gemma-2-2B | 16379 | N/A | Expressions of sadness and emotional distress | "sad" | 1 |
| Gemma-2-2B | – | N/A | – | "mania" | 1 |
| Gemma-2-2B | – | N/A | – | "manic" | 1 |
| Gemma-2-2B | – | N/A | – | "paranoia" | 1 |
| Gemma-2-2B | – | N/A | – | "paranoid" | 1 |

| | | | | | |
|---|---|---|---|---|---|
| Gemma-2-2B | 1659 | N/A | References to suicide and related themes | "suicide" | 2 |
| Gemma-2-2B | 13207 | N/A | Terms related to suicide and mental health conditions | "suicide" | 2 |
| Gemma-2-2B | 6983 | N/A | Emotional expressions of sadness or grief | "sad" | 2 |
| Gemma-2-2B | 2137 | N/A | Expressions of sadness and related emotions | "sad" | 2 |
| Gemma-2-2B | – | N/A | – | "mania" | 2 |
| Gemma-2-2B | – | N/A | – | "manic" | 2 |
| Gemma-2-2B | – | N/A | – | "paranoia" | 2 |
| Gemma-2-2B | – | N/A | – | "paranoid" | 2 |
| Gemma-2-2B | 431 | N/A | Instances of suicide and related topics | "suicide" | 3 |
| Gemma-2-2B | 880 | N/A | Expressions of sadness and related emotions | "sad" | 3 |
| Gemma-2-2B | 3809 | N/A | Emotional references to tears and related expressions of sadness or distress | "sad" | 3 |
| Gemma-2-2B | – | N/A | – | "mania" | 3 |
| Gemma-2-2B | – | N/A | – | "manic" | 3 |

| Gemma-2-2B | – | N/A | – | "paranoia" | 3 |
|---|---|---|---|---|---|
| Gemma-2-2B | – | N/A | – | "paranoid" | 3 |
| Gemma-2-2B | – | N/A | – | "suicide" | 4 |
| Gemma-2-2B | – | N/A | – | "sad" | 4 |
| Gemma-2-2B | – | N/A | – | "mania" | 4 |
| Gemma-2-2B | – | N/A | – | "manic" | 4 |
| Gemma-2-2B | – | N/A | – | "paranoia" | 4 |
| Gemma-2-2B | – | N/A | – | "paranoid" | 4 |
| Gemma-2-2B | 4174 | N/A | Terms related to assisted and unassisted suicide, particularly in the context of medical care and decision-making | "suicide" | 5 |
| Gemma-2-2B | – | N/A | – | "sad" | 5 |
| Gemma-2-2B | – | N/A | – | "mania" | 5 |
| Gemma-2-2B | – | N/A | – | "manic" | 5 |
| Gemma-2-2B | – | N/A | – | "paranoia" | 5 |
| Gemma-2-2B | – | N/A | – | "paranoid" | 5 |
| Gemma-2-2B | 5991 | N/A | References to suicide and self-harm | "suicide" | 6 |
| Gemma-2-2B | 4948 | N/A | Expressions and sentiments related to sadness and emotional distress | "sad" | 6 |

| | | | | | |
|---|---|---|---|---|---|
| Gemma-2-2B | 520 | N/A | Expressions of sadness and emotional distress | "sad" | 6 |
| Gemma-2-2B | 14567 | N/A | Emoticons or symbols representing emotions, particularly sadness or disappointment | "sad" | 6 |
| Gemma-2-2B | – | N/A | – | "mania" | 6 |
| Gemma-2-2B | – | N/A | – | "manic" | 6 |
| Gemma-2-2B | – | N/A | – | "paranoia" | 6 |
| Gemma-2-2B | – | N/A | – | "paranoid" | 6 |
| Gemma-2-2B | 14326 | N/A | Words and phrases related to suicide and self-harm | "suicide" | 7 |
| Gemma-2-2B | 8820 | N/A | Expressions of sadness or disappointment | "sad" | 7 |
| Gemma-2-2B | – | N/A | – | "mania" | 7 |
| Gemma-2-2B | – | N/A | – | "manic" | 7 |
| Gemma-2-2B | – | N/A | – | "paranoia" | 7 |
| Gemma-2-2B | – | N/A | – | "paranoid" | 7 |
| Gemma-2-2B | 3258 | N/A | Terms and phrases related to capital punishment and suicide | "suicide" | 8 |

| | | | | | |
|---|---|---|---|---|---|
| Gemma-2-2B | 10938 | N/A | Concepts related to suicide and depression | "suicide" | 8 |
| Gemma-2-2B | 3813 | N/A | Expressions of sadness and negative emotions | "sad" | 8 |
| Gemma-2-2B | 5031 | N/A | Expressions of sadness and crying | "sad" | 8 |
| Gemma-2-2B | – | N/A | – | "mania" | 8 |
| Gemma-2-2B | – | N/A | – | "manic" | 8 |
| Gemma-2-2B | – | N/A | – | "paranoia" | 8 |
| Gemma-2-2B | – | N/A | – | "paranoid" | 8 |
| Gemma-2-2B | 11149 | N/A | References to self-harm and suicide | "suicide" | 9 |
| Gemma-2-2B | 7042 | N/A | Expressions of sadness and disappointment | "sad" | 9 |
| Gemma-2-2B | – | N/A | – | "mania" | 9 |
| Gemma-2-2B | – | N/A | – | "manic" | 9 |
| Gemma-2-2B | – | N/A | – | "paranoia" | 9 |
| Gemma-2-2B | – | N/A | – | "paranoid" | 9 |
| Gemma-2-2B | 5154 | N/A | Terms related to murder and suicide | "suicide" | 10 |
| Gemma-2-2B | 6381 | N/A | Expressions of emotional distress and sadness | "sad" | 10 |

| | | | | | |
|---|---|---|---|---|---|
| Gemma-2-2B | 9071 | N/A | Expressions and descriptions of sadness or negative emotional states | "sad" | 10 |
| Gemma-2-2B | 13526 | N/A | Emotions related to laughter and humor in contrast to expressions of sadness or seriousness | "sad" | 10 |
| Gemma-2-2B | 16347 | N/A | Expressions of emotional impact and resonance within narratives, particularly those that evoke strong feelings like sadness and humanity | "sad" | 10 |
| Gemma-2-2B | – | N/A | – | "mania" | 10 |
| Gemma-2-2B | – | N/A | – | "manic" | 10 |
| Gemma-2-2B | – | N/A | – | "paranoia" | 10 |
| Gemma-2-2B | – | N/A | – | "paranoid" | 10 |
| Gemma-2-2B | 1776 | N/A | Phrases related to suicide and the circumstances surrounding it | "suicide" | 11 |

| | | | | | |
|---|---|---|---|---|---|
| Gemma-2-2B | 9840 | N/A | Expressions of sadness and disappointment | "sad" | 11 |
| Gemma-2-2B | 16356 | N/A | Expressions of sadness and emotional distress | "sad" | 11 |
| Gemma-2-2B | – | N/A | – | "mania" | 11 |
| Gemma-2-2B | – | N/A | – | "manic" | 11 |
| Gemma-2-2B | – | N/A | – | "paranoia" | 11 |
| Gemma-2-2B | – | N/A | – | "paranoid" | 11 |
| Gemma-2-2B | 6552 | N/A | Discussions and terms related to reproductive rights and legal implications surrounding abortion and suicide | "suicide" | 12 |
| Gemma-2-2B | 9907 | N/A | Content related to death and suicide | "suicide" | 12 |
| Gemma-2-2B | 11281 | N/A | Expressions of disappointment and sadness | "sad" | 12 |
| Gemma-2-2B | – | N/A | – | "mania" | 12 |
| Gemma-2-2B | – | N/A | – | "manic" | 12 |
| Gemma-2-2B | – | N/A | – | "paranoia" | 12 |
| Gemma-2-2B | – | N/A | – | "paranoid" | 12 |

| | | | | | | |
|---|---|---|---|---|---|---|
| Gemma-2-2B | 15292 | N/A | Themes related to suicide and self-sacrifice | "suicide" | 13 |
| Gemma-2-2B | 1655 | N/A | Emotions and personal expressions related to feelings of sadness and reflection | "sad" | 13 |
| Gemma-2-2B | 5037 | N/A | Expressions of sadness or emotional distress | "sad" | 13 |
| Gemma-2-2B | – | N/A | – | "mania" | 13 |
| Gemma-2-2B | – | N/A | – | "manic" | 13 |
| Gemma-2-2B | – | N/A | – | "paranoia" | 13 |
| Gemma-2-2B | – | N/A | – | "paranoid" | 13 |
| Gemma-2-2B | 11259 | N/A | References to specific individuals and personal circumstances related to loss and suicide | "suicide" | 14 |
| Gemma-2-2B | 12019 | N/A | Subjects related to suicide and its implications | "suicide" | 14 |
| Gemma-2-2B | 6673 | N/A | Expressions of sadness and emotional pain | "sad" | 14 |
| Gemma-2-2B | – | N/A | – | "mania" | 14 |
| Gemma-2-2B | – | N/A | – | "manic" | 14 |
| Gemma-2-2B | – | N/A | – | "paranoid" | 14 |

| Gemma-2-2B | – | N/A | – | "paranoia" | 14 |
|---|---|---|---|---|---|
| Gemma-2-2B | 132 | N/A | Mentions of a specific character or brand, particularly related to the context of suicide | "suicide" | 15 |
| Gemma-2-2B | 4657 | N/A | Expressions of sadness or negative emotions | "sad" | 15 |
| Gemma-2-2B | – | N/A | – | "mania" | 15 |
| Gemma-2-2B | – | N/A | – | "manic" | 15 |
| Gemma-2-2B | – | N/A | – | "paranoia" | 15 |
| Gemma-2-2B | – | N/A | – | "paranoid" | 15 |
| Gemma-2-2B | – | N/A | – | "suicide" | 16 |
| Gemma-2-2B | 1383 | N/A | Expressions of sadness and disappointment | "sad" | 16 |
| Gemma-2-2B | – | N/A | – | "mania" | 16 |
| Gemma-2-2B | – | N/A | – | "manic" | 16 |
| Gemma-2-2B | – | N/A | – | "paranoia" | 16 |
| Gemma-2-2B | – | N/A | – | "paranoid" | 16 |
| Gemma-2-2B | – | N/A | – | "suicide" | 17 |
| Gemma-2-2B | 1010 | N/A | Expressions of sadness or negative emotions | "sad" | 17 |

| | | | | | |
|---|---|---|---|---|---|
| Gemma-2-2B | 4018 | N/A | Emotional states and their implications, particularly focusing on sentiments of fear, sadness, and resignation in various context | "sad" | 17 |
| Gemma-2-2B | 8273 | N/A | Expressions of emotions, particularly sadness and personal reflections | "sad" | 17 |
| Gemma-2-2B | – | N/A | – | "mania" | 17 |
| Gemma-2-2B | – | N/A | – | "manic" | 17 |
| Gemma-2-2B | – | N/A | – | "paranoia" | 17 |
| Gemma-2-2B | – | N/A | – | "paranoid" | 17 |
| Gemma-2-2B | 9168 | N/A | Phrases related to suicide and self-harm | "suicide" | 18 |
| Gemma-2-2B | 7236 | N/A | Emotive expressions related to loss and sadness | "sad" | 18 |
| Gemma-2-2B | 12844 | N/A | Emotional expressions and reactions, particularly those reflecting disappointment or sadness | "sad" | 18 |

| | | | | | |
|---|---|---|---|---|---|
| Gemma-2-2B | 2433 | N/A | Negative emotions or sentiments, particularly in relation to distress or sadness | "sad" | 18 |
| Gemma-2-2B | 11186 | N/A | Emotional expressions related to loss and sadness | "sad" | 18 |
| Gemma-2-2B | – | N/A | – | "mania" | 18 |
| Gemma-2-2B | – | N/A | – | "manic" | 18 |
| Gemma-2-2B | – | N/A | – | "paranoia" | 18 |
| Gemma-2-2B | – | N/A | – | "paranoid" | 18 |
| Gemma-2-2B | 8943 | N/A | Words and phrases related to suicide and self-harm | "suicide" | 19 |
| Gemma-2-2B | 1068 | N/A | Emotional expressions related to crying and sadness | "sad" | 19 |
| Gemma-2-2B | – | N/A | – | "mania" | 19 |
| Gemma-2-2B | – | N/A | – | "manic" | 19 |
| Gemma-2-2B | – | N/A | – | "paranoia" | 19 |
| Gemma-2-2B | – | N/A | – | "paranoid" | 19 |
| Gemma-2-2B | 7098 | N/A | References to suicide and self-harm | "suicide" | 20 |
| Gemma-2-2B | 534 | N/A | References to death and suicide-related themes | "suicide" | 20 |

| | | | | | |
|---|---|---|---|---|---|
| Gemma-2-2B | 15539 | N/A | Expressions of disappointment or sadness | "sad" | 20 |
| Gemma-2-2B | 15682 | N/A | Expressions of sadness and mourning | "sad" | 20 |
| Gemma-2-2B | – | N/A | – | "mania" | 20 |
| Gemma-2-2B | – | N/A | – | "manic" | 20 |
| Gemma-2-2B | – | N/A | – | "paranoia" | 20 |
| Gemma-2-2B | – | N/A | – | "paranoid" | 20 |
| Gemma-2-2B | 4087 | N/A | Phrases related to suicide and self-harm | "suicide" | 21 |
| Gemma-2-2B | 15760 | N/A | Expressions of sadness and cruelty | "sad" | 21 |
| Gemma-2-2B | 4801 | N/A | Expressions of disappointment or sadness | "sad" | 21 |
| Gemma-2-2B | 11505 | N/A | Expressions of sadness and emotional turmoil | "sad" | 21 |
| Gemma-2-2B | – | N/A | – | "mania" | 21 |
| Gemma-2-2B | – | N/A | – | "manic" | 21 |
| Gemma-2-2B | – | N/A | – | "paranoia" | 21 |
| Gemma-2-2B | – | N/A | – | "paranoid" | 21 |
| Gemma-2-2B | – | N/A | – | "suicide" | 22 |
| Gemma-2-2B | 8680 | N/A | Japanese phrases expressing sadness or despair | "sad" | 22 |
| Gemma-2-2B | – | N/A | – | "mania" | 22 |

| | | | | | |
|---|---|---|---|---|---|
| Gemma-2-2B | – | N/A | – | "manic" | 22 |
| Gemma-2-2B | – | N/A | – | "paranoia" | 22 |
| Gemma-2-2B | – | N/A | – | "paranoid" | 22 |
| Gemma-2-2B | 13998 | N/A | Japanese characters and words related to serious topics such as suicide and mental health issues | "suicide" | 23 |
| Gemma-2-2B | 1279 | N/A | Elements related to self-harm and suicide methods | "suicide" | 23 |
| Gemma-2-2B | 5943 | N/A | Terms related to suicide attempts and self-harm | "suicide" | 23 |
| Gemma-2-2B | – | N/A | – | "sad" | 23 |
| Gemma-2-2B | – | N/A | – | "mania" | 23 |
| Gemma-2-2B | – | N/A | – | "manic" | 23 |
| Gemma-2-2B | – | N/A | – | "paranoia" | 23 |
| Gemma-2-2B | – | N/A | – | "paranoid" | 23 |
| Gemma-2-2B | – | N/A | – | "suicide" | 24 |
| Gemma-2-2B | – | N/A | – | "sad" | 24 |
| Gemma-2-2B | – | N/A | – | "mania" | 24 |
| Gemma-2-2B | – | N/A | – | "manic" | 24 |
| Gemma-2-2B | – | N/A | – | "paranoia" | 24 |
| Gemma-2-2B | – | N/A | – | "paranoid" | 24 |

| | | | | | |
|---|---|---|---|---|---|
| Gemma-2-2B | 8626 | N/A | Topics related to suicide and mental health crises | "suicide" | 25 |
| Gemma-2-2B | 15435 | N/A | Terms related to suicide, particularly methods and contexts | "suicide" | 25 |
| Gemma-2-2B | 1305 | N/A | Data related to suicide statistics and occurrences | "suicide" | 25 |
| Gemma-2-2B | 14170 | N/A | References to mental health and suicide | "suicide" | 25 |
| Gemma-2-2B | – | N/A | – | "sad" | 25 |
| Gemma-2-2B | – | N/A | – | "mania" | 25 |
| Gemma-2-2B | – | N/A | – | "manic" | 25 |
| Gemma-2-2B | – | N/A | – | "paranoia" | 25 |
| Gemma-2-2B | – | N/A | – | "paranoid" | 25 |