# OpenReview forum: "pSAE-chiatry: Utilizing Sparse Autoencoders to Uncover Mental-Health-Related Features in Language Models"
_ICLR.cc/2025/Conference — Submitted to ICLR 2025_

### Official Review · Reviewer_PTwS · 2024-11-01

**Soundness:** 2
**Presentation:** 3
**Contribution:** 2
**Rating:** 5
**Confidence:** 3

**Summary:**

This paper examines the relationships between suicide, mania, and psychosis by analyzing the internal representations of Mental Health-Related Features (MHRF) in the Gemma-2-2B language model. The study identified MHRF across all 25 layers of the model, discovering 29 features related to suicide and 42 related to sadness. However, no features associated with mania or paranoia were found, indicating potential gaps in the model's ability to address complex psychiatric symptoms. Additionally, the research revealed that prompts related to suicide activated a newly identified, suicide-related feature more strongly. These findings highlight the need for better identification and modulation of features within AI models to improve their safety and effectiveness in mental health care applications.

**Strengths:**

1. This paper examines the relationships between suicide, mania, and psychosis by analyzing the internal representations of the MHRF. It identifies MHRF across all 25 layers of the model and discovers 29 features related to suicide and 42 features related to sadness.
2. The paper also finds that no features associated with mania or paranoia were identified, indicating potential gaps in the model's ability to address complex psychiatric symptoms.
3. These findings highlight the need for improved identification and modulation of features within AI models to enhance their safety and effectiveness in mental health care applications.

**Weaknesses:**

1. The reasons why focusing on suicide, mania, and psychosis instead of other MHRF is not fully emphasized.
2. More statistical analysis can be performed to better uncover the relationships and show the statistical significance of the correlation.

**Questions:**

1. More comprehensive statistical analysis should be performed.
2. More ethical review is needed.

---

### Official Review · Reviewer_vEgr · 2024-11-03

**Soundness:** 1
**Presentation:** 2
**Contribution:** 1
**Rating:** 1
**Confidence:** 5

**Summary:**

This paper investigates mental health-related features in the Gemma-2-2B language model using GemmaScope's sparse autoencoders on Neuronpedia. The authors search for specific terms related to mental health crises (suicide, mania, paranoia) and look for feature activation patterns using a limited set of prompts.

**Strengths:**

1. This addresses an important problem in AI safety for mental health applications.
2. Provides initial exploration of mental health-related features in language models
3. Evaluated SAE's in a new domain

**Weaknesses:**

Fundamental Methodological Issues:
- No developments on existing methodology - simply searching GemmaScope on Neuronpedia for specific terms. Comparing multiple SAE’s e.g. pretrained vs instruction tuned versions or training new SAE’s would be a useful avenue to consider.
- Relies entirely on GPT-4 generated feature labels without validating if these labels accurately represent the underlying features. It would be more useful to use saelens to calculate the activations across a variety of approaches and evaluate activation patterns.
- No discussion or justification of activation thresholds, which is critical for determining feature relevance, further max pooling vs single activation was not explored. This should be conducted with saelens to get activations for each token in input and comparing how aggregation methods impacts what is considered activated or not- there is little consensus on true standards at present and thus the sensitivity analysis needs to be done.
- Extremely limited evaluation (only four prompts) without statistical significance


Technical Limitations:
- Single model setup without comparison across sizes, width, sae location— see saelens and neuronpedia for available pretrained location and widths
- No correlation analysis between feature activation and model outputs ie just because the sae latent activates does not mean this will be used in the output i.e. no systematic evaluation of activation patterns' usefulness for screening
- Missing analysis of false positives/negatives in feature detection in this output space


Limited Scope:
- Surface-level analysis of a single model
- No exploration of feature interactions or complex patterns
- Missing systematic prompt engineering to test feature robustness
- No comparison with baseline or other methods e.g probing methods or random controls

**Questions:**

Statistical Validation & Thresholds:
- Could you conduct a systematic analysis of different activation thresholds (e.g., testing values from 0.1 to 0.9) and quantify how they impact feature identification?
- What statistical tests could you employ to validate that the activation differences between suicide-related and homicide-related prompts are significant?

Feature Validation:
- Could you create a validation dataset of 20-50+ diverse prompts (including negative examples) to test the robustness of your identified features?
- Would you consider doing ablation studies where you systematically mask each identified feature to measure its causal impact on model outputs?
- How would you validate that these features capture clinically relevant concepts beyond just keyword matching?

Experimental Design:
- Could you expand your analysis to include at least 3 different widths and model sizes and evaluate how this affects feature detection?

Clinical Relevance:
- Would you consider creating a test set of real crisis conversations (appropriately anonymized) to validate feature detection in realistic scenarios?
- How would you measure false positive/negative rates for crisis detection based on feature activation?

---

### Official Review · Reviewer_Rp6o · 2024-11-04

**Soundness:** 1
**Presentation:** 2
**Contribution:** 1
**Rating:** 1
**Confidence:** 5

**Summary:**

This manuscript aims to understand the internal representations of mental health-related variables in Language Models. The manuscript focuses on Gemma 2B and utilize an existing service- Gemma Scope to probe and analyze variables of interest.

**Strengths:**

The manuscript utilizes existing methods to study biases in LLMs (Gemma 2-2B). I believe the strength of the manuscript is the psychiatry-based perspective it offers on text generated by LLMs and their hidden representations.

**Weaknesses:**

- While the subject of understanding the limitations of LLMs' biases/limitations in generating content that has less harm to the user is interesting, the manuscript's contributions/findings are preliminary. The methodologies proposed lack novelty but help build a case for more work. There is no mathematical framework proposed but an existing software or API.
- It seems like the manuscript may benefit from a more in-depth technical and mathematical understanding of why LLMs generate such content.
- The content of the paper stops at finding that 'MHRFs' pertaining to a subset of variables were identified but others could be probed. May be there are subsequent questions that follow that finding?
- This manuscript may find a better audience in psychiatry or other domain-focused venues.
- The methods used are limited in scope and rely on an existing service. Are there methods that are generalizable from the perspective of both psychiatry and Machine learning/NLP?

**Questions:**

While the content of the manuscript is fine as is for the goals it set out, I believe there is a lot more that could be done.
- What are the possible next steps to expand these analyses?
- Can or are there mathematical frameworks that could be applied to provide possible solutions to the problems highlighted?
- Given the findings highlighted in the paper (only a subset of MHRFs), what are some solutions?
- What are some ways to provide human feedback to models to ensure they are more aligned to the goals that were built for?
- On the other hand, what are some possible strengths of using these models in clinical and healthcare situations? For e.g., do they help diagnose a patient with mental health conditions swiftly? Do they reduce clinical burden on physicians or cause unintended biases?

---

### Official Review · Reviewer_VvgR · 2024-11-04

**Soundness:** 1
**Presentation:** 1
**Contribution:** 2
**Rating:** 3
**Confidence:** 3

**Summary:**

This paper explores how the Gemma-2-2B language model internally represents mental health crises, like suicidality and sadness, using sparse autoencoders and insights from mental health professionals. The researchers found that while the model has several features tied to sadness and suicide, it doesn’t recognize more complex issues like mania or paranoia, which could limit its effectiveness in handling certain mental health scenarios. By testing how the model responds to different prompts, they show that adjusting these features can actually change the model’s behavior, pointing to potential ways to make AI responses safer and more sensitive in mental health settings.

**Strengths:**

Originality: This paper makes a meaningful and original contribution by applying sparse autoencoders to uncover latent features in language models specifically related to mental health crises.

Significance: The author addresses an important and underexplored topic, potentially making a significant impact.

**Weaknesses:**

1: The paper’s structure makes it challenging to follow the main points and understand its contributions. There’s no clear section that lays out the contributions upfront, so readers have to dig through the text to find them. Important findings, like the model’s lack of features for recognizing mania and paranoia, aren’t highlighted enough, which lessens their impact.

2. In the introduction, the problem is mentioned in the first paragraph, but the proposed solution only appears at the very end, with little explanation connecting the two. Adding a brief, clear link between the problem and solution would help readers see how analyzing model features could address the issues the paper raises. This would make the overall purpose and approach easier to follow.

3. The choice of Gemma-2-2B as the model for this study is insufficiently justified, with the authors only briefly mentioning in the last paragraph of the methods section that it was chosen due to its accessibility, low cost, and manageable computational demands on Neuronpedia. This explanation, however, doesn’t address why Gemma-2-2B is an appropriate or optimal choice for investigating mental health-related features. Without any comparison to other models, it’s unclear if the findings—such as the absence of features for mania or paranoia—are unique to Gemma-2-2B or indicative of broader trends across language models.

4. The authors select specific terms—“suicide,” “sad,” “mania,” “manic,” “paranoia,” and “paranoid”—to search for mental-health-related features in the model, yet they do not provide any theoretical foundation, clinical criteria, or systematic reasoning behind this choice. This omission raises questions about the comprehensiveness and relevance of the chosen terms for representing complex psychiatric conditions.

5. One of the paper’s most important findings—that the model has 42 features related to “sad” and 29 features related to “suicide” distributed across its 25 layers—is not emphasized adequately in the main results section. Instead, this key data is relegated to the appendix, where it still lacks clear academic or statistical presentation.

**Questions:**

Could you elaborate on why Gemma-2-2B was selected as the model for this study? Beyond its accessibility and low cost, are there specific characteristics that make it suitable for identifying mental health-related features?

What criteria guided the selection of terms like “suicide,” “sad,” “mania,” “manic,” “paranoia,” and “paranoid” as representative of mental health features?

You mention filtering out irrelevant features (e.g., “Wrestlemania” and “Romania” in searches for “mania”). Could you clarify how irrelevant terms were identified and excluded? Was this done manually or through an automated process, and were there specific criteria to determine relevance?

---

### Meta-Review · Area_Chair_sdB6 · 2024-12-16

**Metareview:**

This paper applies tools from mechanistic interpretability to LLMs (or, more specifically, Gemma-2-2b) in the context of mental health applications. Specifically, using sparse auto-encoders, they identify "mental health features" within the model hidden states, and highlight that these do not fully cover mental health problems.

There was consensus amongst reviewers that this paper offers some interesting (though preliminary) results, and that the application area considered is important. However, there was also agreement that the work seems insufficiently framed and motivated, and the experimental setup is limited and inadequately justified. Moreover, there is really no technical contribution here.

**Additional Comments On Reviewer Discussion:**

The authors did not respond to the reviews.

---

### Decision · Program_Chairs · 2025-01-22

Reject